# Dexamethasone Induces the Expression and Function of Tryptophan-2-3-Dioxygenase in SK-MEL-28 Melanoma Cells

**DOI:** 10.3390/ph14030211

**Published:** 2021-03-04

**Authors:** Marta Cecchi, Sara Paccosi, Angela Silvano, Ali Hussein Eid, Astrid Parenti

**Affiliations:** 1Department of Health Sciences, Clinical Pharmacology and Oncology Section, University of Florence, Viale Pieraccini 6, 50139 Florence, Italy; marta.cecchi@unifi.it (M.C.); sara.paccosi@unifi.it (S.P.); angela.silvano@unifi.it (A.S.); 2Department of Basic Medical Sciences, College of Medicine, QU Health, Qatar University, Doha P.O. Box 2713, Qatar; 3Biomedical and Pharmaceutical Research Unit, QU Health, Qatar University, Doha P.O. Box 2713, Qatar

**Keywords:** SK-Mel-28, melanoma, tryptophan-2,3-dioxygenase, indoleamine-2,3-dioxygenase-1, dexamethasone, 680C91, epacadostat, migration, proliferation, MMP2

## Abstract

Tryptophan-2,3-dioxygenase (TDO) is one of the key tryptophan-catabolizing enzymes with immunoregulatory properties in cancer. Contrary to expectation, clinical trials showed that inhibitors of the ubiquitously expressed enzyme, indoleamine-2,3-dioxygenase-1 (IDO1), do not provide benefits in melanoma patients. This prompted the hypothesis that TDO may be a more attractive target. Because the promoter of TDO harbors glucocorticoid response elements (GREs), we aimed to assess whether dexamethasone (dex), a commonly used glucocorticoid, modulates TDO expression by means of RT-PCR and immunofluorescence and function by assessing cell proliferation and migration as well as metalloproteinase activity. Our results show that, in SK-Mel-28 melanoma cells, dex up-regulated TDO and its downstream effector aryl hydrocarbon receptor (AHR) but not IDO1. Furthermore, dex stimulated cellular proliferation and migration and potentiated MMP2 activity. These effects were inhibited by the selective TDO inhibitor 680C91 and enhanced by IDO1 inhibitors. Taken together, our results demonstrate that the metastatic melanoma cell line SK-Mel-28 possesses a functional TDO which can also modulate cancer cell phenotype directly rather than through immune suppression. Thus, TDO appears to be a promising, tractable target in the management or the treatment of melanoma progression.

## 1. Introduction

L-tryptophan (Trp) is an essential amino acid that plays important roles in protein synthesis as well as the biosynthesis of melatonin, serotonin, and nicotinamide adenine dinucleotide (NAD^+^) [1]. The role of Trp catabolism in cancer biology has been receiving increased interest due to its implication in cancer immune evasion [2]. Indoleamine-2,3-dioxygenase-1 (IDO1) and tryptophan-2,3-dioxygenase (TDO) are the main enzymes within the first and rate-limiting step of the kynurenine pathway (KP) of Trp catabolism. Most Trp are catabolized by IDO1, which is ubiquitously expressed and can be induced by interferon gamma (IFN-γ) [3]. Interestingly, IDO1 has been widely demonstrated to possess immunosuppressive functions, which are established to correlate with poor survival in various cancer patients [2] (Figure 1).

Despite all efforts of early diagnosis, metastatic melanoma continues to have a poor prognosis [4]. In recent years, a better understanding of the role of the immune system in cancer has led to the approval of several immunotherapies using monoclonal antibodies against immune checkpoints, such as ipilimumab (anti CTLA-4) as well as nivolumab and pembrolizumab (anti PD-1) [4]. Unfortunately, only specific subgroups of patients responded to these immunotherapies, pointing out IDO1 as a further ideal candidate [3]. It was indeed reported that reduced serum tryptophan concentrations could be predictive markers for melanoma [5]. Although selective and potent IDO1 inhibitors have shown promising results in experimental models of cancer [6], their benefit in melanoma patients has not been completely elucidated. Indeed, in a phase 3 randomized, double blind study, ECHO-301/KEYNOTE-252, a selective IDO1 inhibitor, epacadostat 100 mg twice daily, plus pembrolizumab did not improve progression-free survival or overall survival compared with placebo plus pembrolizumab in patients with unresectable or metastatic melanoma [7]. Due to the uncertain benefit of IDO1 inhibition as a strategy to enhance immune checkpoints activity, and since approximately 35% of tumor cancer cell lines express TDO [8], this enzyme became a very attractive target in cancer immunotherapy [9].

TDO is physiologically expressed in the liver where it plays a critical role in regulating Trp levels. Following some stimuli, however, TDO’s expression can be detected in several organs, including testis [10], placenta [11], and brain [12]. More recent studies have pointed out the relevance of TDO in some cancers, namely malignant glioma, melanoma, bladder cancer, and triple-negative breast carcinoma [13,14]. Indeed, TDO appears to be constitutively expressed in these cancer cells, and its upregulation has been intimately associated with the ability of tumor cells to evade immune surveillance [15]. Therefore, TDO may represent an attractive target, especially when IDO1 does not account for constitutive Trp catabolism [8,16].

The glucocorticoid (GC) derivative dexamethasone (dex) is routinely used as a co-medication in cancer therapy to ameliorate some side effects of chemotherapeutic agents [17,18]. In hematological malignancies, particularly in multiple myeloma, dex is part of all chemotherapy protocols, owing to its strong apoptosis-inducing effects [19]. However, GCs could also inhibit chemotherapy-induced tumor cell’s apoptosis in in vitro and in vivo experimental models as well as in freshly obtained surgical specimens of some tumors, including melanoma [20].

The effects of dex on tumor cell growth remain inconclusive, likely due to the many factors involved, including dose and cell context [21,22]. Therefore, to clearly determine the clinical relevance of GCs, their effects on the malignant phenotype ought to be elucidated. Since TDO promotor possesses glucocorticoid response elements (GREs) [23], a role for TDO in tumor biology and progression has attracted attention, especially in the context of melanoma.

Based on these considerations, this study was undertaken to better characterize TDO/KP in SK-Mel-28, a human melanoma cell line. We previously demonstrated that TDO is constitutively expressed in these cells and that it directly regulates their proliferation [24]. However, the effect of dex on TDO expression and function remained unclear. Here, we seek to determine this role in order to better delineate glucocorticoid’s effects on melanoma cells.

## 2. Results

### 2.1. Dexamethasone Increased TDO and AHR Expression

Since TDO promotor harbors GREs, we wished to determine the effect of dexamethasone (dex) on its expression in SK-Mel-28 cells. Our results show that dex indeed increases TDO mRNA (TDO2) expression in concentration (data not shown) and time-dependent manners. The maximal effect was observed after six hours of treatment with 25 µM of dex (Figure 2A). Immunofluorescence analysis further confirmed the increase in protein levels of TDO in response to dex (Figure 2B).

Given the influence of IDO1 on melanoma’s malignant phenotype, we next aimed to determine its expression in SK-Mel-28. Electrophoresis of the amplified and purified target show that IFN-γ (50 ng/mL) but not dex (25 µM) increased expression of IDO1 (Figure 3A). This was further confirmed by real-time PCR (Figure 3B) and immunofluorescence (Figure 3C). Interestingly, although IDO1 mRNA levels were almost undetectable, an IDO inhibitor, 1-MT, significantly enhanced dex-induced TDO2 upregulation (Figure 3D). This may suggest that SK-Mel-28 cells could use TDO as principal enzyme for the activation of the KP.

Among the downstream effectors of KP, the aryl hydrocarbon receptor (AHR) represents an important target. Our results show that treatment with dex upregulated mRNA and protein levels of AHR (Figure 4A,B). Interestingly, this effect was rapid, as it was significantly increased within 3 h of stimulation with dex (Figure 4A). This increase is not sustained since the level of AHR after 24 h of dex does not seem to cause an appreciable change over basal level (Figure 4B). This is likely due to the fact that dex accelerates both ligand-dependent and ligand independent AHR protein degradation in a GR-dependent manner [25].

### 2.2. Dexamethasone Stimulates SK-Mel-28 Proliferation via TDO and PI3K/Akt

Because TDO is involved in the regulation of SK-Mel-28 growth [24], and since dex induced TDO up-regulation, we then wished to determine whether dex stimulates SK-Mel-28 proliferation via TDO. Our results show that dex significantly and concentration-dependently increased SK-Mel-28 proliferation (Figure 5A). The maximal effect was obtained with 25 µM, which caused a 48.4 ± 9% increase in cell duplication compared to control unstimulated cells. Dex-promoted cell proliferation appears to be dependent on glucocorticoid receptors and on increased transcriptional activity, since it was abolished by the steroid receptor antagonist RU486 (1 µM) and by actinomycin-D (10 nM), a DNA-dependent RNA synthesis inhibitor (Figure 5B).

To delineate the mechanisms underlying the proliferative effect of dex, cells were pretreated for 15 min with 680C91 (40 μM), a selective TDO inhibitor, followed by dex (25 μM). Interestingly, 680C91 significantly hampered cell proliferation by 41.2 ± 7.1%, without affecting either cell viability [24] or cell duplication in response to FGF2 (Figure 5C). Furthermore, inhibition of IDO1 with 1-MT or epacadostat, a newer IDO1 inhibitor, potentiated dex-induced cell proliferation (Figure 5D).

It is well-documented that dex activates mitogenic pathways such as the Akt and the MAPK cascades in normal [26,27] and tumor cells [21]. Here, we assessed whether dex indeed activates these pathways. Treatment with 25 μM dex stimulated Akt phosphorylation within 3 h, and this effect was partially but significantly inhibited by 680C91 (Figure 6A,B), while ERK1/2 was not activated (data not shown). The dex-activated PI3K/Akt pathway appears to mediate cell proliferation, since inhibiting this pathway did impair SK-Mel-28 proliferation. Consistently, U0126, an MEK inhibitor, did not significantly alter cell proliferation (Figure 6C).

### 2.3. Dexamethasone Stimulates SK-Mel-28 Migration

Wound healing assay was employed to examine the effect of dex on cell migration. Our results show that, compared to control cells (1% fetal bovine serum (FBS)), dex-stimulated cells exhibited significantly higher migratory capacity (Figure 7A,B). This effect was further potentiated by epacadostat, an IDO1 inhibitor. Furthermore, dex-induced migration was mediated by its cognate receptors since pre-treatment with RU486 abolished dex-induced cell migration. Interestingly, the TDO inhibitor 680C91 delayed dex effect (Figure 7), probably due to dex-induced TDO transcription (Figure 2).

Cell chemotaxis was also assessed by means of the Boyden chamber assay. Data show that dex significantly stimulated SK-Mel-28 cell chemotaxis compared to control unstimulated cells (1% FBS, Figure 7C,D). This effect was further potentiated by epacadostat and was impaired by RU486, actinomycin-D, and the TDO inhibitor 680C91 (Figure 7C,D).

### 2.4. Dexamethasone Effect on MMP2 Activity

Invasion is a hallmark of malignant phenotype of cancer cells, including melanomas. Matrix metalloproteinases (MMPs) are major drivers of cellular invasion. Among these MMPs, MMP2 seems to be the most abundantly secreted by SK-Mel-28 [28]. Thus, we measured MMP2 expression and activity in conditioned media of SK-Mel-28 cells stimulated with dex. Gelatin zymography of control supernatants showed constitutive release of the latent forms of MMP2 visualized as a band at 72 kDa (Figure 8). Dex significantly stimulated the release of MMP2 within 48 h and induced its activation, revealed by the appearance of the band at 62 kDa (Figure 8). This increase in MMP2 activity is significantly diminished by RU486 and by 680C91 (Figure 8). No effect of dex on MMP9 was observed (data not shown).

## 3. Discussion

In addition to playing a key role in the KP, TDO appears to have a vital role in cancer cell’s ability to evade the immune system [29]. TDO harbors GREs in its promoter, thus prompting the speculation that dex and other glucocorticoids (GCs) may regulate its expression and hence may affect tumorigenesis. Indeed, different reports highlighted a role for dex in lymphocytic malignancies as well as epithelial cell-derived cancers. In the former, dex and other GCs are routinely used to induce apoptotic cell death. Conversely, in solid tumors, GCs, mostly dex, are often used at high doses to minimize side effects of chemotherapeutic agents [19,30]. Furthermore, several reports suggest that GCs stimulate expression of anti-apoptotic genes and hence antagonize the ability of cytotoxic drugs to successfully induce cell death [31]. Moreover, in vivo studies on stress-induced GCs suggest a positive relationship between GCs and melanoma progression [32,33].

To this end, we investigated the effects of dex on both TDO expression in SK-Mel-28 cells and on proliferation and migration of these cells. The present paper shows, for the first time, that dex up-regulated both mRNA and protein levels of TDO. Dex also up-regulated AHR, a ligand-activated transcription factor. AHR is activated by KP metabolites, such as kynurenine, and is involved in cell proliferation, inflammation, immunity, and regulation of the melanogenic pathway [34,35,36]. Indeed, AHR diminishes the efficacy of novel immunotherapies by potentiation of the production of antibodies that block immunoregulatory molecules and by suppressing apoptosis of dormant melanoma cells [37]. Moreover, in triple-negative breast cancer cells, the TDO–AHR signaling axis facilitates anoikis resistance and metastasis [13] and stimulates cell migration [38].

Based on these data, it was tempting to investigate the effects of dex on SK-Mel-28 proliferation and migration in order to better assess the potential involvement of TDO in those effects. Here, we showed that dex enhanced proliferative and migratory capacities of SK-Mel-28 cells. We also showed that these effects depend on GR activation, rely on increased transcriptional activity, and are mediated by TDO. We also showed that dex-induced MMP2 activity is also mediated by GR and TDO. Conversely, MMP9 was undetected, in line with previous data [28]. Our results are consistent with previous reports showing that dex and prednisolone stimulate proliferation and survival of many tumor cells, both in vitro and in vivo, by activating GRs and Akt pathway [21]. Contextually, Chaudhuri et al. [39] described a detrimental effect of dex in a young patient with metastatic melanoma. This patient received dex 8 mg/os four times a day for 10 days. Although the patient symptomatically improved, it appears that the subcutaneous metastasis rapidly increased, exacerbating the disease. Moreover, metastatic nodule biopsy demonstrated a specific cytosol receptor for glucocorticoid with a K_D_ of 1.8 × 10^−9^ M. Other evidence confirms that glucocorticoids significantly promote adhesion, migration, invasion of melanoma cells in vitro, and lung metastasis in vivo [40]. These effects were mediated by Akt, ROCK_1/2_, and tissue inhibitors of MMP-2 (TIMP2). In addition, dex is known to promote adhesion and survival of human and murine melanoma cell lines through fibronectin regulation and increased chemo-resistance to anticancer agents such as cisplatin [41]. Indeed, prolonged treatment of melanoma cells with dex results in the formation of a cell subline resistant to dex’s growth inhibitory action by virtue of an acquired phenotype of constitutive activation of PI3K. Not surprisingly, PI3K is considered one of the key factors that regulate cell resistance to dex [42]. Dex was also reported to exhibit antiproliferative and proapoptotic effects in other cells such as HT168 and HT168-M1 cell lines [22]. These seemingly contradictory results may be explained by the fact that higher dex concentrations were used. Nonetheless, no data were indeed reported on possible TDO expression and function on HT168 and HT168-M1 melanoma cell lines. Based on our present results, it is clear that SK-Mel-28 cells physiologically express TDO whose expression and function are modulated by dex.

Since IDO1 is the principal enzyme of the KP in normal and tumor tissues, we studied its expression in SK-Mel-28 in response to dex. However, this melanoma cell line did not show IDO1 mRNA expression in control unstimulated cells, and immunofluorescence showed a faint IDO reactivity which was only upregulated in response to IFN-γ, as reported for other tumor cells [43]. However, when IDO1 was pharmacologically inhibited, dex-induced migration and proliferation were significantly increased, indicating that, in SK-Mel-28, the KP is driven by TDO and may be amplified when IDO1 is impaired. This may be in line with other reports [16] showing that TDO/KP could be involved in cancer biology, particularly when IDO1 does not account for the constitutive Trp catabolism.

Among the possible intracellular pathways activated by dex/TDO in SK-Mel-28, PI3K/Akt seems to be involved in dex-induced phenotypic changes. Indeed, we showed that dex stimulated Akt phosphorylation, which was critical for dex-induced proliferation. This is in line with another study which showed that T-acute lymp hoblastic leukemia (T-ALL) cells acquire resistance to dex-mediated killing through abnormal activation of Akt, resulting in inhibition of the FoxO3a/Bim pathway [44]. Furthermore, dex and prednisolone were reported to increase in vitro survival in 21/65 samples from glucocorticoid-resistant primary leukemias. Importantly, dex-induced proliferation was mediated by PI3K/AKT and p38 mitogen-activated protein kinase [21]. Moreover, dex modulated trastuzumab-induced cell growth inhibition through the restoration of trastuzumab-induced Akt suppression in BT-474 breast cancer cells [45]. All these observations suggest PI3K/Akt as the major pathway involved in pro-survival and proliferative effects of dex.

Therefore, the metastatic melanoma cell line SK-Mel-28 possesses a functional TDO which regulates SK-Mel-28 proliferation and migration in response to dex. It is well known that GCs are able to promote tumor onset and progression by virtue of their systemic immunosuppressive effects. However, GCs can have direct effects on proliferation, migration (our present data), and survival of tumor cells [21]. As shown elsewhere, GCs can also potentiate chemoresistance to therapy through regulation of genes involved in cancer progression [46,47].

Clinical evidence highlights a detrimental effect of systemic steroids during immunotherapy [48]. For instance, it was recently shown that there is an association between immune-related adverse events (irAEs) and outcomes in patients with stage III melanoma treated with adjuvant pembrolizumab within the KEYNOTE-054 trial [49]. Similarly, use of systemic corticosteroids during anti-PD1 nivolumab therapy was associated with poorer outcomes in non-small-cell lung cancer patients [48,50,51]. Hence, targeting glucocorticoids to increase cancer immunotherapy efficacy is an intriguing strategy. In this context, it is important to note that glucocorticoid-induced tumor necrosis factor receptor-related protein (GITR) could be an additional therapeutic target involving corticosteroid pathway. GITR is a receptor on T cells capable of inhibiting T cell receptor-induced apoptosis [52]; its natural ligand, GITRL, is part of the TNF superfamily, and it is expressed by a variety of cells, including mature and immature dendritic cells [52,53]. The GITR-GITRL interplay can reverse the suppression by T_reg_ cells while stimulating effector T cells [52,54]. Therefore, modulation of GITR is involved in the anti-inflammatory action of corticosteroids, making it a therapeutic target in immune regulation [52]. Preliminary results of a humanized monoclonal antibody (TRX518) that triggers GITR, given as monotherapy to patients with refractory solid tumors (including melanoma), demonstrated that TRX518 reduces circulating and intratumoral T_reg_ cells to a similar extent, providing an easily evaluable biomarker of anti-GITR activity [55]. More recently, an open-label, phase 1/2a study evaluated the effect of GITR-agonist IgG1 monoclonal antibody, BMS-986156, with or without nivolumab, on 292 patients with advanced solid tumors [56]. Results of this trial showed that BMS-986156 exhibits a manageable safety profile, and its combination with nivolumab is safe and efficacious, much like nivolumab monotherapy. Despite these encouraging data, further evidence in this field is still needed.

In conclusion, given the uncertain efficacy of IDO1 inhibitors together with immune checkpoint inhibitors in advanced melanoma and, based on our current results, demonstrating a direct effect of dex on TDO expression and function in human SK-Mel-28 melanoma cell line, it becomes of increasing importance to better characterize the involvement of the TDO pathway in melanoma tumorigenesis. It is also mandatory to elucidate the effects of GCs on solid tumors, including melanoma.

## 4. Materials and Methods

### 4.1. Cell Culture

SK-Mel-28 (ATCC, Manassas, VA, USA), a human metastatic melanoma cell line, was grown in high D-glucose DMEM, with 10% (*v*/*v*) heat inactivated fetal bovine serum (FBS Defined Hyclone; ThermoFisher Scientific, Waltman, Massachusetts, USA), 100 U/mL penicillin, 100 μg/mL streptomycin, and 2 mmol/L glutamine in a humidified atmosphere with 5% CO_2_ in air. The culture medium was changed every 2 days.

### 4.2. RT-PCR and Real Time PCR

Total RNA was isolated using TRI Reagent and quantified spectroscopically with a NanoDrop (ThermoFisher Scientific). One μg of RNA was used for the reverse transcription reaction with Prime Script RT reagent Kit with gDNA eraser (Takara, Otsu, Japan), and the cDNA samples obtained were amplified with specific primers described below.

Qualitative PCR for IDO1 was performed using Wonder Taq Thermostable DNA polymerase (EuroClone, Milan, Italy). Primers for IDO1 were: fw 5′-AGTTCTGGGATGCATCACCA-3′and rev 5′-TGATCGTGGATTTGGTGAAA-3′. PCR amplification comrpised 40 cycles of initial denaturation at 95 °C for 1 min, denaturation at 95 °C for 15 s, annealing at 53 °C for 15 s, and extension at 72 °C for 30 s, followed by 5 min at 72 °C, using 2720 Thermal Cycler Applied Biosystem (ThermoFisher Scientific). Due to undetectable cDNAs, they were concentrated combining four samples (800 ng), purified with QIAquick PCR Purification Kit (Qiagen, Hilden, Germany) according to the manufacturer instructions, and separated by 1.8% agarose electrophoresis.

Quantitative real-time PCR (qRT-PCR) was carried out using SYBR Premix Ex Taq (Takara) according to the manufacturer instructions on a Rotorgene RG-3000A cycle system (Qiagen) platform. Primer sequences were the following: TDO2 fw: 5′-CTTATCTCCAGCATCAGGCTTCCAGAGT-3′ and rev: 5′-GGAGTTCTTTCCAGCCATGCCTCC-3′ [24], IDO1 amplification fw: 5′-AGTTCTGGGATGCATCACCA-3′ and rev: 5′-CAGTTTCTTGGAGAGTTGGCA-3′ and AHR amplification fw: 5′-CAAATCCTTCCAAGCGGCATA-3′ and rev: 5′-CGCTGAGCCTAAGAACTGAAA-3′.

qRT-PCR amplification of 18 s ribosomal mRNA was used as a normalizer. 18 s amplification fw: 5′-ATTAAGGGTGTGGGCCGAAG-3′ and rev: 5′-GGTGATCACACGTTCCACCT-3′.

The cycle was set at 95 °C for 5 s, 55 °C for 30 s, and 72 °C for 30 s, repeated 35 times. Quantitative real-time polymerase chain reaction data analysis was accomplished with delta delta CT method.

### 4.3. Cell Proliferation

Cell proliferation was quantified by total DNA/well via a fluorescent dye (Cell proliferation kit, Invitrogen, ThermoFisher Scientific) [57]. Briefly, cells were plated on flat-bottom 96-multiwell plates and allowed to adhere overnight. Following a 24 h starving condition, cells were stimulated with increasing dex concentrations with or without several inhibitors: the TDO selective inhibitor, 680C91 (40 μM), or the MEK inhibitor, UO126 (10 μM), or the PI3K inhibitor, LY294002 (10 μM). After 48 h, 50 μL of dye binding solution were added to each microplate well and incubated at 37 °C for 45 min. Fluorescence intensity was read using a fluorescence microplate reader with excitation at ~485 nm and emission detection at ~530 nm.

### 4.4. Chemotactic Assay

Cell migration was assessed with the modified Boyden chamber (48-multiwell plates; Neuroprobe, Gaithersburg, MD, USA) as previously reported [58]. Briefly, 1% fetal calf serum (FCS)-containing medium alone (control) or supplemented with 25 μM dex was added to the lower wells, while 20 × 10^4^ cells were seeded into the upper wells without or with RU486, 680C91, or actinomycin D and incubated at 37 °C for 24 h. Methanol-fixed cells were stained with Diff-Quik (Dade Behring, Dudingen, Switzerland), and cell migration was measured by microscopic evaluation of the number of cells moved across the filter in ten randomly selected fields at magnification 400×. Each experimental point was measured in triplicate.

### 4.5. Wound-Healing Assay

Cells were cultured in 24-well plates, and after they reached 95% confluence, a straight cell-free wound was made by manual scratching using a sterile 1000 μL pipette tip. Cells were washed with PBS twice to eliminate any cellular debris, and a fresh medium was added, which served as the negative control (1% FBS, control time 0 h). Cells were then stimulated with 25 μM dex alone or in the presence of inhibitors. After 24 h, cells migrated into the wounded area were visualized and photographed using an inverted light microscope. The area covered by cells was calculated with Image J (version 1.46) software, and results are reported as percent of migration rate, calculated using the formula:Migration rate (%) = (Area time 0 h − Area time 24 h/Area time 0 h) × 100.

### 4.6. Western Blotting Analysis

SK-Mel-28 cells were lysed in a Triton^®^ X-100 lysis buffer, pH 7.4, followed by a centrifugation at 14,000× *g* rpm for 10 min at 4 °C [57]. Cell lysate was run on 10% SDS-polyacrylamide gel electrophoresis, blotted onto PVDF membrane (Merck-Millipore, Darmstadt, Germany), and immunostained with anti-phospho Akt (1:1000; Cell Signaling Technology, MA, USA), anti-AHR (1:1000; Cell Signaling Technology), and anti β-tubulin monoclonal antibody (1:1000; Sigma-Aldrich, MO, USA). The antigen–antibody complexes were visualized using appropriate secondary antibodies and the ECL detection system by means of ChemiDoc imaging system (Bio-Rad Laboratories, Hercules, CA, USA)

### 4.7. Immunofluorescence

SK-Mel-28 were seeded (1 × 10^4^ cells) in high D-glucose DMEM with 10% FBS onto LabTek Slides Chamber and incubated in a 5% CO_2_ atmosphere at 37 °C for 24 h. Then, a fresh medium with low serum concentration (1% FBS) was added, and cells were stimulated with dex or IFN-γ for 3, 6, and 24 h. Double immunofluorescence analyses were performed on cells after fixation with cold acetone for 5 min. Nonspecific binding sites were blocked with 10 mg/mL bovine serum albumin in PBS for 1 h at room temperature with 0.2% triton X-100 (Sigma-Aldrich), then were treated with a primary antibody, i.e., monoclonal rabbit anti-human AhR (1: 100; Cell Signaling Technology), polyclonal rabbit anti-human IDO1 (1:200; abcam), or monoclonal anti human-TDO (1:200; Novus Biologicals) overnight at 4 °C, then treated for 2 h at room temperature with a secondary goat anti-rabbit or anti-mouse antibodies conjugated with FITC AF488 (green fluorescence), all from Life Technology (Thermo Fisher Scientific). The signal was amplified with anti-FITC fluorescein/Oregon green antibody for 1.5 h (1:100; Invitrogen) at room temperature. The nuclei were labeled with Hoechst 33,342 (20 μg/mL; Sigma; blue fluorescence). Omission of primary antibodies was used as negative controls. The slides were mounted with Fluoromount and examined with Leica DC200 microscope digital color camera and Leica DC Viewer software.

### 4.8. Gelatin Zymography

MMP-2 and MMP-9 activity was assessed as gel zymography [57]. Starved cells were stimulated with 25 µM dex in the presence or the absence of 680C91 (40 μM) and epacadostat (1 μM). After 24 h, the media were collected, clarified by centrifugation at 14,000× *g* RPM for 7 min, and subjected to electrophoresis onto 8% SDS-PAGE containing 1 mg/mL gelatin under non-denaturing conditions. Following electrophoresis, gels were washed with 2.5% Triton X-100 and incubated for 48 h at 37 °C in a 50 mM Tris buffer containing 200 mM NaCl and 20 mM CaCl_2_, pH 7.4. Gels were then stained with 0.5% Coomassie brilliant blue R-250 in 10% acetic acid and 45% methanol and destained with 10% acetic acid and 45% methanol, all from Sigma-Aldrich. Gelatinase activity was then evaluated by quantitative densitometry.

### 4.9. Materials

The 680C91 ((E)-6-fluoro-3-[2-(3-pyridyl)vinyl]-1H-indole), 1-MT, a competitive inhibitor of IDO1, anti β-tubulin monoclonal antibody, TRI Reagent, and actinomycin-D were from Merck (KGaA, Darmstadt, Germany). Mouse monoclonal anti-TDO antibody was from NovusBio (Bio-Techne, Minneapolis, MI, USA). high D-glucose DMEM, and PBS were from Euroclone S.p.A. (Pero, Milan, Italy). Epacadostat was from DivBioScience, Ulvenhout, The Netherlands.

### 4.10. Statistical Evaluation

Statistical analysis was performed with Prism software (GraphPad 5.02). Parametric data were reported as means ± SEM, and differences between groups were tested with ANOVA test (followed by Bonferroni’s and Dunnett’s Multiple Comparison Test) as appropriate. Alpha value was set at 0.05.

## Figures and Tables

**Figure 1 pharmaceuticals-14-00211-f001:**
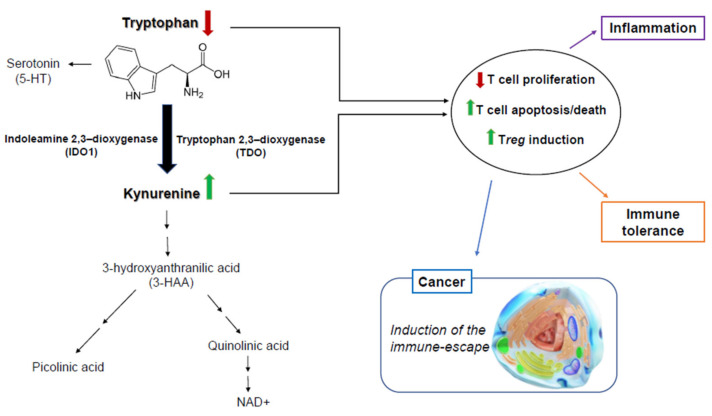
Tryptophan catabolism and kynurenine pathway. Indoleamine-2,3-dioxygenase (IDO1) and tryptophan-2,3-dioxygenase (TDO) pathways control T cell and T_reg_ responses. IDO1—and TDO but to a lesser extent—are expressed on professional antigen presenting cells and tumor cells and are critical in immune regulation of cancers, infections, and inflammation.

**Figure 2 pharmaceuticals-14-00211-f002:**
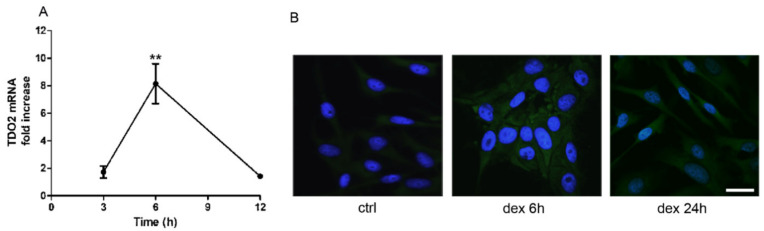
Dexamethasone (dex) increases mRNA and protein expression of TDO. (**A**) Cells were treated with dex (25 µM) for different time points. RT-PCR shows time-dependent increase in mRNA of TDO2. Data are plotted as mean ± SEM (*n* = 8, ** *p* < 0.01 vs. control). (**B**) Cells treated without (ctrl) or with dex (25 µM) for 6 or 24 h, then subjected to immunofluorescence, TDO (green fluorescence), nuclei (blue). Representative photomicrographs at 40× magnification are shown. Scale bar 20 µm.

**Figure 3 pharmaceuticals-14-00211-f003:**
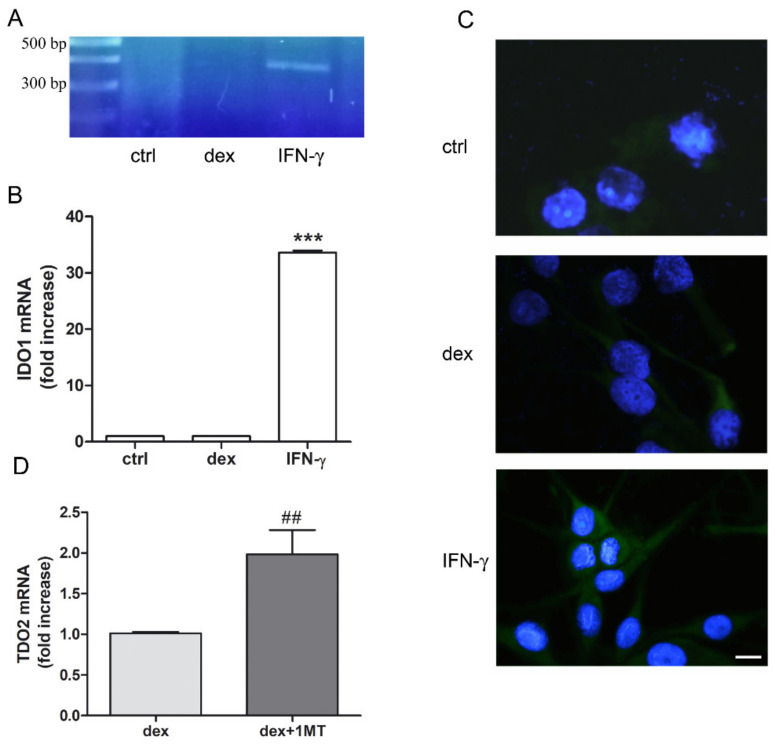
Modulation of IDO1 and TDO2 by interferon gamma (IFN-γ) or dex. (**A**) Cells were treated without (ctrl) or with dex (25 µM) or with IFN-γ (50 ng/mL) for 24 h. Total RNA was then isolated, and RT-PCR-generated cDNA of IDO1 was subjected to agarose gel electrophoresis. (**B**) Cells were treated without (ctrl) or with dex (25 µM) or with IFN-γ (50 ng/mL) for 24 h. Total RNA was then isolated, and real time PCR was performed. Data represent mean ± SEM, (*n* = 4; *** *p* < 0.001 vs. control unstimulated (ctrl) cells). (**C**) Cells were treated without (ctrl) or with dex (25 µM) or with IFN-γ (50 ng/mL) for 24 h, followed by immunofluorescence for IDO1 expression (IDO1: green; nuclei: blue). Representative photomicrographs at 40× magnification are shown. Scale bar 20 µm. (**D**) Cells were treated with dex (25 µM) in the presence or absence of 40 µM 1-MT, an IDO inhibitor. Real-time PCR results for TDO2 expression are shown. Data plotted represent mean ± SEM. Increase over dex effect (*n* = 6; ## *p* < 0.01 vs. dex alone).

**Figure 4 pharmaceuticals-14-00211-f004:**
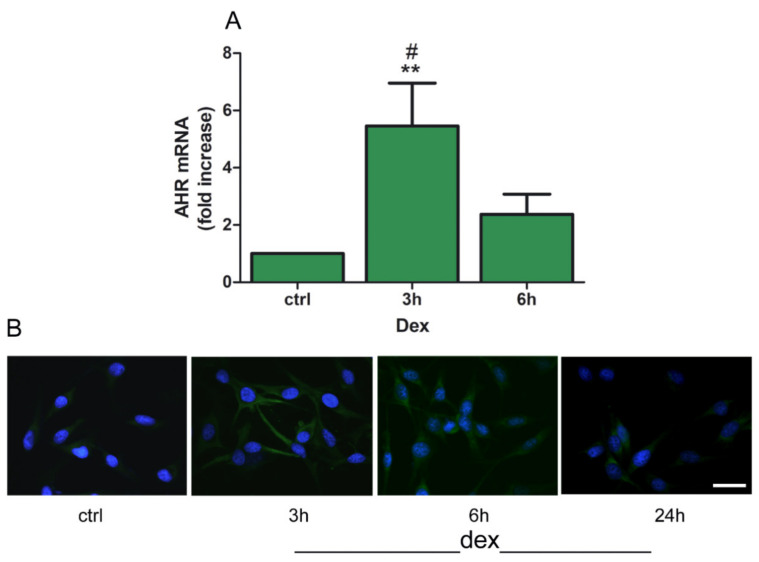
Dex modulates aryl hydrocarbon receptor (AHR) expression. (**A**) Cells were treated without (ctrl) or with dex (25 µM) for 3 and 6 h. Total RNA was then isolated, and real time PCR was performed. Data represent mean ± SEM, (*n* = 5; ** *p* < 0.001 vs. control unstimulated (ctrl) cells; # *p* < 0.05 vs. dex 3 h. (**B**) Cells were treated without (ctrl) or with dex (25 µM) for 3, 6, and 24 h followed by immunofluorescence for AHR expression (AHR: green; nuclei: blue). Representative photomicrographs at 40× magnification are shown. Scale bar 20 µm.

**Figure 5 pharmaceuticals-14-00211-f005:**
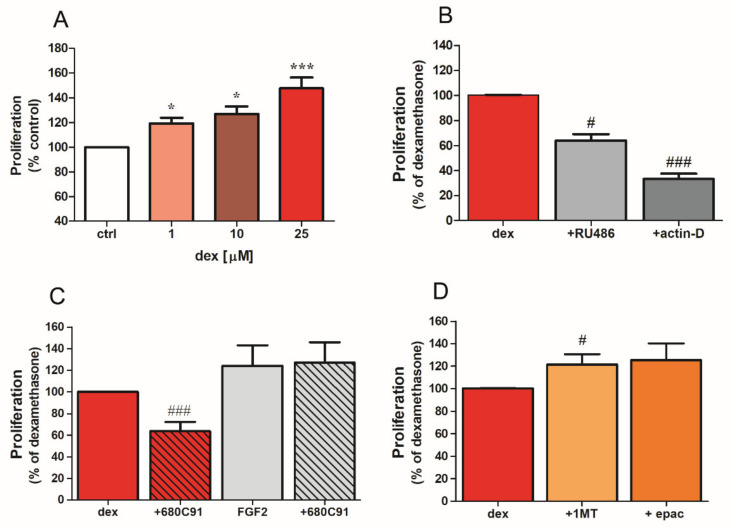
TDO mediates dex-induced SK-Mel-28 proliferation. (**A**) Cells were treated without (ctrl) or with increasing concentrations of dex for 48 h. Data are plotted as percent of control. Mean ± SE (*n* = 7; * *p* < 0.05, *** *p* < 0.001 versus ctrl). (**B**–**D**) Cells were treated with dex (25 μM) alone (dex) or in presence of (**B**) RU486 (1 µM) or actinomycin D (10 nM) or (**C**) 680C91 (40 µM) and FGF2 (10 ng/mL) or (**D**) 1-MT (40 µM) or epacadostat (1 µM), and proliferation was assessed. Data are plotted as mean ± SE (*n* = 7; # *p* < 0.05, ### *p* < 0.001 vs. dex alone).

**Figure 6 pharmaceuticals-14-00211-f006:**
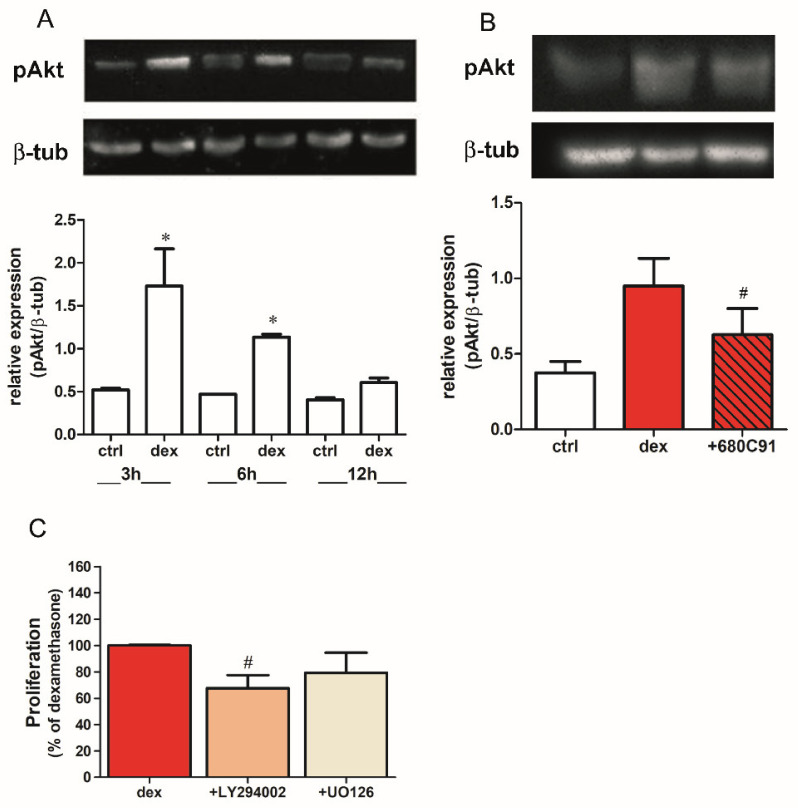
Dex activates Akt in SK-Mel-28 cells. (**A**) Cells were treated with dex for different durations (3, 6, or 12 h). Western blotting for phosphorylated Akt and β-tubulin. (**B**) Cells were stimulated with 25 µM dex alone (dex) or in presence of 40 µM 680C91 (+680C91) for 3 h (* *p* < 0.05, vs. ctrl; # *p* < 0.05 vs. dex alone). (**C**) Cells were treated with dex alone (dex) or co-treated with LY294002 (5 µM) or U0126 (10 µM), Akt and MEK inhibitors, respectively, and proliferation assessed. Data are plotted as mean ± SE (*n* = 3; # *p* < 0.05, vs. dex alone).

**Figure 7 pharmaceuticals-14-00211-f007:**
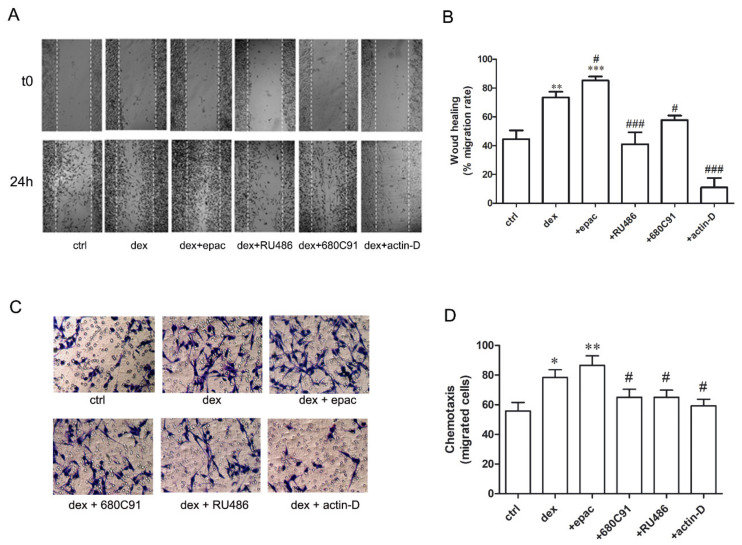
Dex stimulates SK-Mel-28 cell migration. (**A**,**B**) Scratched monolayer was treated with 25 µM dex for 24 h in absence (dex) or in presence of RU486 (1 µM), 680C91 (40 µM), epacadostat (1 µM), or actinomycin-D (10 Nm). (**A**) Photomicrographs of the wound were taken, and representative images are shown (4×). (**B**) Quantitative measure of the wound’s width. Data represent mean ± SEM of the migration rate (*n* = 5; ** *p* < 0.01, *** *p* < 0.001 vs. control unstimulated cells (ctrl); # *p* < 0.05, ### *p* < 0.001 vs. dex alone. (**C**,**D**) Chemotaxis of SK-Mel-28 cells stimulated with 25 µM dex for 24 h in absence (dex) or in presence of RU486 (1 µM), 680C91 (40 µM), epacadostat (1 µM) or actinomycin-D (10 nM). (**C**) Photomicrographs of the migrated cells were taken, and representative images are shown (20×). (**B**) Quantitative measure of cell chemotaxis. Data represent mean ± SEM of migrated cells (*n* = 4; * *p* < 0.05, ** *p* < 0.01 vs. control unstimulated cells (ctrl); # *p* < 0.05 vs. dex alone.

**Figure 8 pharmaceuticals-14-00211-f008:**
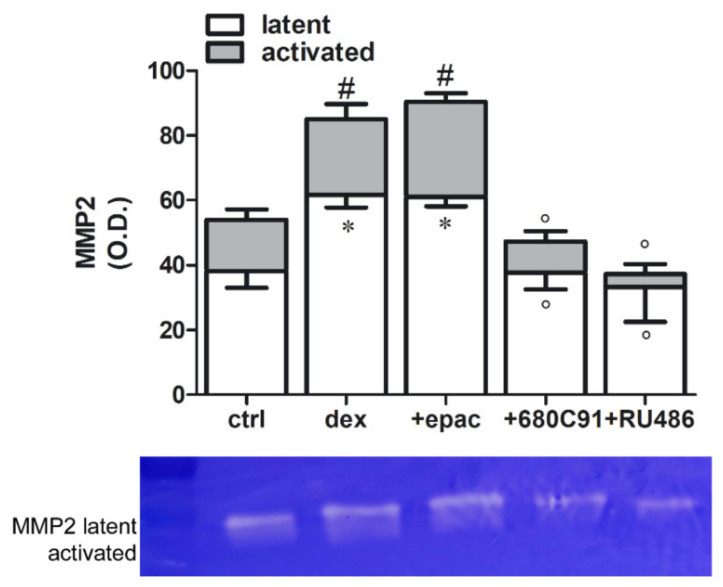
Gelatin zymography for matrix metalloproteinases (MMP2) production in response to dex alone or in presence of epacadostat (1 µM) or 680C91 (40 µM) or RU486 (1 µM). Results are reported as mean ± SEM of band densitometry, *n* = 5. * *p* < 0.05 vs. latent MMP2 and # *p* < 0.05 vs. activated MMP2 of control unstimulated cells (ctrl); ° *p* < 0.05 vs. dex alone.

## Data Availability

Data are available at the Dept of Health Sciences, University of Florence, Viale Pieraccini 6, 50139 Florence, Italy.

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
