# Peer review of "Dexamethasone Induces the Expression and Function of Tryptophan-2-3-Dioxygenase in SK-MEL-28 Melanoma Cells"

_pharmaceuticals, 2021, doi:10.3390/ph14030211_

Round 1

Reviewer 1 Report

This is an interesting and well written manuscript with conclusions consistent with the experimental data. The clinical translational impact of the study  shloud be better explained in the discussion section.

Reviewer 2 Report

The article makes a good impression and it is interesting in that it contributes to the study of the relationship between glucocorticoids and melanoma progression. The results of the work show the role of tryptophan-2,3-dioxygenase (TDO), the key tryptophan-catabolizing enzymes with immunoregulatory properties in cancer, in a possible mechanism.

The work was done at a good level, however, a minor question still arises:

  • the authors demonstrate that dexamethasone stimulated Akt phosphorylation. Perhaps the data would be better understood if densitometry data were presented.

The results obtained indicate the necessity and prospects for the further study of TDO-dependent pathway in the tumorigenesis of melanoma.

Reviewer 3 Report

authors describe in-vitro assays in a melanoma cell-line to investigate if dexamethasone modulates Tryptophan-2,3-dioxygenase expression and alters cellular functions / properties as migration and proliferation. Nicely written manuscript, igures could use some improvements (resolution too low?, or its just the pdf)

Major comment:

All experiments have been done on one cell-line andmight not reflect this cell-line and not melanoma in general. Also: there is no evidence given this relates to the human situation.

Introduction:

The introduction would benefit from an overvieuw /image of the relevant pathways 

Methods/ results:

Wound-healing assay: The confluency of 95% is rather unspecific and cells as well as the surface of the plate is damaged when making the "wound". Why did the authors choose this assay over a ring barrier assay and how was 95% confluency measured. In figure 6 A there are clear differences in cell number outside of the "wound". How does this all realte to the specific counts generated and visualized in figure 1B? 

Results: it might be the pdf, but I do not/hardly see the gereen signal in the fluorescence images. Also, in the legend of these figures it should be mentioned what is stained with (not only in the methods). Figures should be interpretable as is, not by going back to the methods.

Discussion

the discussion lacks sufficient discussions in associating dexametasone use in patients and the effect on relevant pathways investigated.

Reviewer 4 Report

Thanks for the opportunity to review this manuscript. In this paper the Authors present data on a novel potential therapeutic target with immunomodulatory properties, the Tryptophan-2,3-dioxygenase (TDO). Results of this study suggest that metastatic melanoma cells possess a functional TDO which can modulate cancer cell phenotype, thus playing an immunomodulatory effect.

The article is interesting and worthy of publication, however some minor revisions are required. I would also suggest to widen the discussion to include some relevant issues on this topic (see further).

Comments:

Introduction:

- L-Tryptophan (Trp) is an essential amino acid that plays important roles in protein 33 synthesis as well as the biosynthesis of melatonin, serotonin, and nicotinamide adenine 34 dinucleotide (NAD+). Please add reference for this.

- please add explanation for TDO and IDO (explained in the abstract, should be provided also the first time the term is used in the manuscript).

- The Authors should provide at least a minimal background on the role of immunotherapy in melanoma, in order to provide a correct context for research on the immunomodulatory property of TDO.

- Lines 41-42: the Authors state that “Although selective and potent IDO1 41 inhibitors have shown promising results in experimental models of cancer, their benefit in melanoma patients has not completely reported”. In this regard, the Authors should elaborate the results of the KEYNOTE-252 trial: Epacadostat 100 mg twice daily plus pembrolizumab did not improve progression-free survival or overall survival compared with placebo plus pembrolizumab in patients with unresectable or metastatic melanoma. The usefulness of IDO1 inhibition as a strategy to enhance anti-PD-1 therapy activity in cancer remains uncertain (Long GV, Dummer R, Hamid O, et al. Epacadostat plus pembrolizumab versus placebo plus pembrolizumab in patients with unresectable or metastatic melanoma (ECHO-301/KEYNOTE-252): a phase 3, randomised, double-blind study. Lancet Oncol. 2019;20(8):1083-1097. doi: 10.1016/S1470-2045(19)30274-8).

-Lines 46-47: Please provide reference for: “More recent studies have pointed out the relevance of TDO in some cancers, namely malignant glioma, melanoma, bladder cancer and triple-negative breast carcinoma”.

Results:

-Line 84: “Given the influence of IDO1 on melanoma’s malignant phenotype, we next to determine its expression in SK-Mel-28”, I think this sentence should be rephrased (maybe a verb is missing after “next”?).

Discussion:

-Lines 196-199: please provide reference for: “Indeed, different reports highlighted a role for dex in lymphocytic malignancies as well as epithelial cell-derived cancers. In the for mer, dex and other GCs are routinely used to induce apoptotic cell death. Conversely, in solid tumors, where GCs, mostly dex, are often used at high doses to minimize side effects of chemotherapeutic agents.”.

-It is worthy to add the following topics to the discussion: the role of the glucocorticoid-induced tumor necrosis factor receptor-related protein (GITR) and preliminary results of anti-GITR in association with immunotherapy in metastatic melanoma, as a potential additional therapeutic target involving corticosteroid pathway and metabolism; I would also cite the potential detrimental effect of corticosteroids for the treatment of immune-related adverse events, on the efficacy of immunotherapy (for both topics please see Indini A, Rijavec E, Grossi F. Immune related adverse events and response to immunotherapy: Focus on corticosteroids. Lung Cancer. 2020;145:225. doi: 10.1016/j.lungcan.2020.02.009).

Round 2

Reviewer 3 Report

Authors jave addressed the concerns / suggestions sufficiently. I have no further comments